# Metabolomic Analysis Reveals the Mechanisms of Hepatotoxicity Induced by Aflatoxin M1 and Ochratoxin A

**DOI:** 10.3390/toxins14020141

**Published:** 2022-02-15

**Authors:** Ya-Nan Gao, Chen-Qing Wu, Jia-Qi Wang, Nan Zheng

**Affiliations:** 1Key Laboratory of Quality & Safety Control for Milk and Dairy Products of Ministry of Agriculture and Rural Affairs, Institute of Animal Sciences, Chinese Academy of Agricultural Sciences, Beijing 100193, China; gyn758521@126.com (Y.-N.G.); wuchenqing111@163.com (C.-Q.W.); jiaqiwang@vip.163.com (J.-Q.W.); 2Laboratory of Quality and Safety Risk Assessment for Dairy Products of Ministry of Agriculture and Rural Affairs, Institute of Animal Sciences, Chinese Academy of Agricultural Sciences, Beijing 100193, China; 3Milk and Milk Products Inspection Center of Ministry of Agriculture and Rural Affairs, Institute of Animal Sciences, Chinese Academy of Agricultural Sciences, Beijing 100193, China; 4State Key Laboratory of Animal Nutrition, Institute of Animal Sciences, Chinese Academy of Agricultural Sciences, Beijing 100193, China

**Keywords:** aflatoxin M1, ochratoxin A, hepatoxicity, metabolomics, LysoPCs

## Abstract

Aflatoxin M1 (AFM1) is the only toxin with the maximum residue limit in milk, and ochratoxin A (OTA) represents a common toxin in cereals foods. It is common to find the co-occurrence of these two toxins in the environment. However, the interactive effect of these toxins on hepatoxicity and underlying mechanisms is still unclear. The liver and serum metabolomics in mice exposed to individual AFM1 at 3.5 mg/kg b.w., OTA at 3.5 mg/kg b.w., and their combination for 35 days were conducted based on the UPLC-MS method in the present study. Subsequent metabolome on human hepatocellular liver carcinoma (Hep G2) cells was conducted to narrow down the key metabolites. The phenotypic results on liver weight and serum indicators, such as total bilirubin and glutamyltransferase, showed that the combined toxins had more serious adverse effects than an individual one, indicating that the combined AFM1 and OTA displayed synergistic effects on liver damage. Through the metabolic analysis in liver and serum, we found that (i) a synergistic effect was exerted in the combined toxins, because the number of differentially expressed metabolites on combination treatment was higher than the individual toxins, (ii) OTA played a dominant role in the hepatoxicity induced by the combination of AFM1, and OTA and (iii) lysophosphatidylcholines (LysoPCs), more especially, LysoPC (16:1), were identified as the metabolites most affected by AFM1 and OTA. These findings provided a new insight for identifying the potential biomarkers for the hepatoxicity of AFM1 and OTA.

## 1. Introduction

It has been reported that 25% of agricultural products are contaminated by various mycotoxins [1]. In addition, mycotoxins also occur in some certain indoor spaces such as agricultural product processing places and moldy buildings [2]. Aflatoxin M1 (AFM1) is the only mycotoxin with maximum residue limit (MRL) in milk, 0.50 μg/kg set by China and USA and 0.05 μg/kg set by EU. AFM1 has also been re-identified as Group 1 carcinogen [3]. The high occurrence and contamination levels in milk and milk products were reported in South Asians and Africans as more and more attention has been aroused to monitor AFM1 levels [4]. AFM1 contamination conditions in the Chinese market between 2010 and 2016 were also monitored. Although the quality and safety of milk have improved, AFM1 contamination levels in milk samples occasionally exceeded the legal limit set by the EU [5,6,7]. Ochratoxin A (OTA) is commonly found in various raw food commodities [8]. In addition, OTA ranked first in the prevalence of cereals foods based on 9627 samples by meta-analysis [9]. IARC classified OTA as Group 2B carcinogen [10,11]. The co-occurrence of these two toxins has been reported in milk and baby food samples [12,13,14]. On the other hand, AFM1 and OTA also may co-exist in certain organic dust originating from cereal [15]. Given that the co-existed mycotoxins may perform different interactive effects including additive, synergistic, or antagonistic effects, consequently, the potential risk of exposure to both AFM1 and OTA could not be underestimated.

Given that the major target organ of aflatoxins (AFs) is the liver, several studies have demonstrated its liver toxicity. A meta-analysis showed that the increased risk of liver cirrhosis is associated with AFs exposure [16]. AFB1 significantly resulted in acute liver injury in rats, including hepatic histopathological damage and abnormal liver function [17]. It has been demonstrated that AFB1-induced hepatotoxicity via oxidative stress and apoptosis or autophagy [18,19]. Although OTA targets the kidney, it also involves liver injury. OTA could result in liver oxidative stress and inflammation in animal models, including mice, duck, and weanling pigs [20,21,22,23].

In this study, we tried to find out the potential mechanisms of hepatoxicity induced by AFM1 and OTA. It has also been reported that metabolomics could demonstrate the metabolic changes and identify the biomarkers [24]. Therefore, metabolomics was conducted in the model of CD-1 mice and HepG2 cells. These findings will provide new insights into combined toxic effects of different mycotoxins in daily foods and the environment.

## 2. Results

### 2.1. AFM1 and OTA-Induced Liver Injury in Mice 

The bodyweight of mice was recorded over 35 days (Figure 1A). Compared with the control group, there was no significant difference in the DMSO vehicle group of final weight (*p* > 0.05), while individual AFM1 and OTA as well as their combination significantly decreased the weight (*p* < 0.05, Figure 1B). The liver weight was significantly decreased in the combination of AFM1 and OTA group (*p* < 0.05, Figure 1C), while no significant difference was shown in the liver index (*p* > 0.05, Figure 1D). The serum indicators related to liver injury (ALT, AST, TBIL, and GCT) were evaluated (Figure 1E–H). Compared with the control group, the level of these four serum biochemical parameters was significantly increased in the combination of the AFM1 and OTA groups (*p* < 0.05). Haematoxylin-eosin (HE) and oil red O (ORO) staining was used to investigate the effects on liver morphology and the extent of lipidation (Figure 2). The staining results of HE showed that toxins, especially for the combined treatment, caused liver damage, including poor cytoplasm, inflammatory infiltration, dilation of sinusoids, and irregular shape of the nucleus (marked by yellow arrows) (Figure 2A). The ORO staining results showed that liver steatosis occurred in the toxin treatment group and the percentage of lipid droplet area in the toxins combination group was significantly increased compared with the control group (*p* < 0.05, Figure 2B,C). 

### 2.2. Multivariate Analysis of the Metabolic Profiles

Metabolomics data of liver and serum was examined through two-dimensional unsupervised PCA to observe the variance and distribution of samples within and between different treatments. The PCA analysis showed that no samples were separated in each group, so all samples were reserved in the subsequent analysis (Figure 3A,C). The supervised OPLS-DA analysis was performed to characterize the metabolite profile in the liver and serum. As illustrated in Figure 3B, DMSO treatment cannot be completely separated from the control group for the liver metabolites, while both individual and combined toxin treated groups were clearly separated from the control group. Similar results were also shown in the serum metabolites (Figure 3D). In addition, both in liver and serum, compared with the AFM1 treatment, OTA treatment was closer to the combined toxins treatment, indicating that OTA exerted a stronger hepatoxicity in mice.

### 2.3. Identification of Differential Metabolites of Mice 

In total, 142 and 169 metabolites were identified by UPLC-MS analysis in mice liver and serum, respectively. The integral metabolic network that contained all identified metabolites in liver and serum among DMSO vehicle, individual, and combined toxins group were depicted in Figure 4 and Figure 5. For the liver, compared with the DMSO vehicle group, the highest number of changed metabolites was present in the AFM1 + OTA group (10 up-regulated and 37 down-regulated), followed by the AFM1 group (13 up-regulated and 9 down-regulated) and OTA group (9 up-regulated and 11 down-regulated) (Figure 4). For serum, the highest number of changed metabolites was also shown in the AFM1 + OTA treatment (20 up-regulated and 28 down-regulated), followed by OTA treatment (29 up-regulated and 14 down-regulated) and AFM1 treatment (5 up-regulated and 27 down-regulated) (Figure 5). The results compared with the control group (Appendix A) were similar to the results compared with the DMSO group. These results suggested that the combined AFM1 and OTA displayed the synergistic effect and OTA might be the one that exerted chief toxic effect.

### 2.4. The Main Type of Metabolites Affected by AFM1 and OTA 

The heatmap analysis was used to visually compare the metabolome levels among samples and treatments (Figure 6). The metabolites in the heatmap were clustered based on their expression levels, and these results depicted that the metabolic patterns among control treatment, DMSO vehicle treatment, and AFM1 treatment were similar, which were significantly different from OTA and AFM1+OTA treatment. These results were consistent with Figure 4, Figure 5 and Figure suggesting that OTA might serve as the predominant toxicity in the combination of AFM1 and OTA. In addition, as the results depicted in Figure 6, among the 20 significant altered metabolites both in liver and serum, LysoPCs were the main type metabolites affected by AFM1 and OTA. Moreover, the concentration of changed LysoPCs in the liver and serum was shown in Appendix A. Furthermore, to more accurately target the differentially expressed metabolites, we conducted the metabolism of HepG2 cells. The concentration of AFM1 (1 μg/mL) and OTA (0.05 μg/mL) was used for metabolome analysis based on cell viability (Appendix A). The metabolites heatmap of HepG2 cells also demonstrated that LysoPCs were involved in the liver toxicity induced by AFM1 and OTA (Appendix A). The expression pattern of LysoPCs in HepG2 cells was similar as in serum (Appendix A). More specially, we found that sn-1 LysoPC (16:1) was the most crucial LysoPCs in the hepatoxicity induced by the combination of AFM1 and OTA, and its concentration in mice and Hep G2 cells is shown in Figure 7.

## 3. Discussion

It is common to find the co-contamination of AFM1 and OTA in the environment. The liver not only plays a vital role in metabolism but also plays a key role in detoxification and excretion of exogenous chemicals [25]. Therefore, it is reasonable to determine the adverse effects caused by AFM1 and OTA on liver tissue and expound on the potential mechanism(s). Given that various interaction types exert in the combined mycotoxins, the interactive effects between AFM1 and OTA also needed to be clarified. In the present study, *in vivo* (CD-1 mice) and *in vitro* (HepG2 cells) models were used and analyzed by metabolomics. Our results showed that AFM1 and OTA led to liver tissue damage, which was associated with LysoPCs metabolites. Moreover, their combination performed a synergistic interactive effect, in which OTA exerted the dominant hepatoxicity role.

The serum biochemical parameters (ALT, AST, TBIL, and GCT) are considered to be indicators for liver impairment. In the present study, AFM1 and its combination with OTA significantly affected the level of these indicators (Figure 1). In normal conditions, ALT and AST mainly present in the liver tissue. When the liver is damaged, part of the ALT and AST in the liver tissue will enter the blood, which will eventually lead to the increase in the activity of ALT and AST in the serum [26]. Broilers exposed to dietary AFB1 caused significantly increased activity of ALT and AST in serum [8]. It has been reported that injured liver and bile duct could lead to increased serum TBIL levels [27]. The increased TBIL concentration was found in the mice fed with 1.5 mg/kg AFB1 for seven days [28]. The level of GCT, a cholestatic liver enzyme, will increase during liver dysfunction [29]. The activity of liver cytosolic GST in broilers exposed to AFB1 was increased for 7, 21, and 42 days [30]. 

The phenotypic results *in vivo* demonstrated that the combined AFM1 and OTA produced more serious adverse effects on liver damage than individually groups (Figure 1 and Figure 2), which indicated that their combination displayed a synergistic effect. Various interactive effects were exerted in combined mycotoxins. The studies regarding the interaction of combined mycotoxins mainly focused on the models *in vitro* [31,32,33]. Only two studies reported the interactive effects of mycotoxins *in vivo*. The interactive effect between AFB1 and deoxynivalenol (DON) was synergism on the injured liver-related parameters including ALT, AST, and albumin (ALB), while the combination of AFB1 and zearalenone (ZEA) produced an antagonistic effect in those parameters [34]. It had been reported that the combination of DON and ZEA displayed an antagonistic effect in the liver tissue of mice [35]. The combined *in vivo* and *in vitro* studies regarding the hepatoxicity of mycotoxins were limited. Only one research demonstrated that synergistic hepatotoxicity was proved in the combined patulin and cadmium *in vivo* (C57BL/6N mice) and *in vitro* (AML12 cells) [36]. Therefore, the *in vivo* results of combined toxins filled the gap to a certain extent.

In the present study, we applied the mice liver and serum metabolomics *in vivo* combined with *in vitro* (HepG2 cells) to explore the underlying mechanisms for the hepatoxicity induced by AFM1 and OTA. The metabolomics of individual liver or serum has been used to characterize toxicological outcomes induced by various hazard factors, such as ethanol and silica nanoparticles [37,38,39]. However, the combined metabolomics of liver and serum is relatively insufficient. Hepatic and serum metabolomics were applied in the toxicity study of individual chemicals of psoralen and bisphenol A [40,41]. More importantly, there were only two studies using metabolomics to expound the interactive effects of combined mycotoxins [35,42]. Therefore, the findings in this study with the combined liver and serum metabolomics on the hepatoxicity interactive effects of AFM1 and OTA could broaden our horizons. The number of differentially expressed metabolites in combined toxins was higher than individual mycotoxins (Figure 4D and Figure 5D), suggesting that the synergistic effect was exerted, which was consistent with the phenotypic results.

In the present study, the metabolomics analysis *in vivo* and *in vitro* provides a possibility to screen the key functional metabolites more accurately, and we found that lysophosphatidylcholines (LysoPCs), more specially sn-1 LysoPC (16:1), played a vital role in the hepatoxicity induced by AFM1 and OTA. There were some discrepancies in the expression pattern of LysoPCs in tissue and HepG2 cells, which could be partially explained by the differences between cells and the whole animal body. Metabolomics analysis demonstrated that the potential biomarkers for hepatocellular carcinoma (HCC) disease were taurocholic acid, lysophosphoethanolamine (LysoPE) 16:0, and LysoPC (22:5) [43]. The analysis of human plasma metabolic profiling suggested that LysoPCs, especially LysoPC (18:1), LysoPC (18:2), LysoPC (20:3), LysoPC (20:4), LysoPC (22:6) were the significantly altered metabolites that have the preventive effect against liver diseases and anti-inflammatory effects [44]. Rats exposed to chronic low-dose acrylamide led to the significant changes of LysoPE (18:3), LysoPE (20:5), and LysoPC (20:4) in the liver [45]. Clinical serum metabolomics demonstrated that the development of liver cirrhosis was related to the levels of metabolites including LysoPC (22:6), myristic acid, and palmitic acid [46].

Phosphatidylcholines (PC) and phosphatidylethanolamines (PE) have been demonstrated to be associated with steatosis and phyospholipidosis [47]. PCs and LysoPCs are negatively correlated, depending on the membrane function synthesis of PCs, and the catalysis of PCs, which could induce lipo-apoptosis [48]. It has been observed that the correlation between acylcarnitine and LysoPC is negatively correlated with PC, assuming that LysoPC is recruited from PC or inhibits PC formation [49]. In this study, serum metabolic profiling showed that the increased level of LysoPCs and decreased level of L-carnitine were performed in toxins treatment. Moreover, the up-regulated LysoPCs could result in cell apoptosis via triggering reactive oxygen species (ROS) production [50,51]. In addition, it has been reported that ROS progress could activate extracellular hydrolyzed by phospholipase A2 (PLA2), and hydrolyze PCs to accumulate LysoPCs [52]. However, during oxidative stress, when LysoPCs combine with acyl groups to form PCs, the down-regulation of LysoPCs may also be related to the toxicity mode [53], which could explain why the concentration of LysoPCs in the liver decreased. L-carnitine has been previously proven to prevent the formation of ROS [54,55]. Accordingly, we speculated that AFM1 and OTA may lead to the oxidative stress of mice liver, inducing the changes in metabolites, eventually damaging the liver. The possible mechanisms of different effects on oxidative damage caused by individual AFM1 and OTA were reported in a previous study [56]. The lipophilic structure of these two toxins and their competition for glutathione in cells could partly explain, the deviations of cytotoxicity induced by AFM1 and OTA, and further studies are needed to research the exact mechanisms. 

Except for hepatotoxicity, the toxicology induced by AFM1 and OTA also had been reported by previous studies. Metabolomics analysis showed that when AFM1 and OTA were combined together, OTA exhibited the dominant effect on the alteration of kidney metabolic processes [57], which was consistent with the role of OTA in the present study. The synergistic effect of the combined AFM1 and OTA was demonstrated in disrupting intestinal integrity, including the decreased cell viability and the expression of tight junction proteins as well as mucins, as well as the increased epithelial permeability and intestinal inflammation [56,58,59,60,61]. The synergy was also consistent with the combined effect on hepatotoxicity induced by AFM1 and OTA in the present study.

In summary, we assess the hepatotoxicity of AFM1 and OTA in the CD-1 mice model. The phenotypic results and metabolomics analysis *in vivo* demonstrated that the combination of AFM1 and OTA performed more serious adverse effects on mice liver, displaying the synergistic effect, in which OTA performed the dominant role. This finding suggested that although the carcinogen classification level of AFM1 (Group 1) is higher than that of OTA (Group 2B), the adverse effects of OTA cannot be ignored. And this finding also provided a basis that OTA should be classified as a higher level of human carcinogen as previous studies demonstrated [10,11]. In addition, the limit standard for OTA in milk should also be considered to establish due to its higher hepatoxicity than AFM1. By means of the combined metabolomics analysis *in vitro* and *in vivo*, we found that LysoPCs, especially, LysoPC (16:1) were the main type of metabolites affected by AFM1 and OTA, which are related to oxidative stress, resulting in the hotoxicity. In order to more accurately understand the mechanism of hepatotoxicity induced by AFM1 and OTA, it would be better to consider combining the obtained metabolomics results with other omics results in the future. 

## 4. Materials and Methods

### 4.1. Chemicals and Reagents

AFM1 and OTA were purchased from Pribolab (Singapore). Alanine aminotransferase (ALT), aspartate aminotransferase (AST), total bilirubin (TBIL), and glutamyltransferase (GCT) assay kit were purchased from Beijing-XinChuangYuan (Beijing, China). Formic acid and acetonitrile (purity > 98% by HPLC) were obtained from Merck (Darmstadt, Germany). The enhanced cell counting kit-8 (CCK-8) was obtained from Beyotime Biotechnology (Shanghai, China). Dulbecco’s modified eagle’s medium (DMEM), fetal bovine serum (FBS), antibiotics (10,000 units/mL penicillin and 10,000 μg/mL streptomycin), and non-essential amino acids solution (NEAA) were purchased from GIBCO (Grand Island, NY, USA).

### 4.2. Animals and Treatment

Sixty CD-1 mice (20 ± 2 g, male) were obtained from Beijing Vital River Laboratory Animal Technology Co., Ltd. (Beijing, China). These mice were fed in the SPF barrier system under the condition of 22 ± 1 °C and 50 ± 5% humidity in China Agricultural University. The animal experiments conducted in the present study were approved by the Ethics Committee of Institute of Animal Sciences, Chinese Academy of Agriculture Sciences (Beijing, China), with the permission code of “IAS2019-3 (Date: 18/3/2019)”.

The mice were randomly separated into five treatments: control treatment (free-feeding), DMSO vehicle treatment (1.0% DMSO), AFM1 treatment (3.5 mg/kg b.w.), OTA treatment (3.5 mg/kg b.w.), and the combination of AFM1 and OTA treatment (3.5 mg/kg b.w. AFM1 + 3.5 mg/kg b.w. OTA). The stock solution of AFM1 and OTA were 700 mg/L and 900 mg/L, respectively, which were dissolved in 0.5% DMSO/ddH_2_O. The dose of AFM1 and OTA used in the present study was demonstrated in our previous study [57]. The mice in DMSO and toxins’ treatment were gavaged once a day (0.2 mL/mice), lasting for 35 days. After treatment, the mice were euthanized by CO_2_. The liver tissue was collected, weighed, and frozen in liquid nitrogen for subsequent histological assessment and metabolomic analysis. 

### 4.3. Determination of Liver Index and Serum Biochemical Indicators

The relative liver weight (liver index) was calculated as the formula: liver index% = liver weight/final weight of the mice ×100. The final weight of the mice was measured before the blood collection. The collection of serum samples was the same as previously reported [62]. The serum biochemical indicators related to liver function, including ALT, AST, TBIL, and GCT were measured following the introductions of enzyme-linked immunosorbent assay (ELISA) kits. 

### 4.4. Histopathological Assessment of Liver

Liver tissues were fixed with 4% paraformaldehyde solution for 24 h, a part of liver samples was dehydrated in alcohol, embedded in paraffin wax with a thickness of 4 mm section, followed by sliced at 5 μm, stained with HE using light microscopy (Nikon Corporation, Tokyo, Japan). Another part of liver tissue was dehydrated in sucrose solution, embedded in optimal cutting temperature compound, followed by being sliced at 8–10 μm and stained with ORO by light microscopy (Nikon Corporation, Tokyo, Japan) to identify the lipid droplets in the tissue. Lipid droplets are orange-red to bright red, and the nucleus are blue. The area of lipid droplets was quantified by Image-Pro Plus 6.0 software, and the percentage of lipid droplets area was calculated as the formula: % lipid droplets area = area of lipid droplets/area of liver tissue ×100.

### 4.5. Cell Culture and Treatment

Human hepatocellular carcinoma cells (HepG2 cells) were obtained from ATCC (Manassas, VA, USA). The cells were cultured in an incubator at 37 °C with 5% CO_2_ with the medium of DMEM supplemented with 10% FBS, 1% antibiotics, and 1% NEAA_._ HepG2 cells were seeded in a 96-well plate at the density of 1 × 10^5^ cells/well and incubated for 24 h. The cells were then treated with AFM1 (0.2, 0.5, 0.8, 1.0, 1.5, 2.0, 3.0, 4.0 and 5.0 μg/mL) and OTA (0.01, 0.02, 0.03, 0.04, 0.05, 0.06, 0.07, 0.08, 0.10, 0.20 and 0.30 μg/mL). The cell viability was measured by CCK-8 assay following the manufacturer’s instructions. Viability (%) = Mean OD in toxin group/Mean OD in control group ×100. 

In the present study, HepG2 cells exposed to 1.0 μg/mL AFM1 and 0.05 μg/mL OTA were used in the subsequent metabolomics analysis. The stock solution of AFM1 and OTA for *in vitro* experiment were 200 μg/mL and 1000 μg/mL, respectively. Therefore, the concentration of DMSO in the AFM1 and OTA used in the study was 0.5% and 0.005%, respectively. 

### 4.6. Metabolomics Analysis

The sample (mice liver, mice serum and HepG2 cells) preparation was conducted. The liver sample in pre-cooled methanol was centrifuged at 1200× *g* for 10 s at 4 °C to obtain the liver supernatant. Centrifuging the serum sample in methanol: water solution at 1200× *g* for 15 min at 4 °C, and then the supernatant of serum was collected. HepG2 cells were incubated at 4 °C with shaking at 150× *g* for 30 min, and then centrifuged at 10,000× *g* for 10 min to collect the supernatant. The collected supernatant of liver, serum, and cell samples were dried in a vacuum evaporator and then detected via LC-MS. Each liver and serum group were composed of ten samples, and each cell group was composed of six samples.

The conditions of UPLC-MS metabolomics assay were reported as our previous study [56]. The instrument parameters were referenced to the previous method [63] and conducted by LipidALL Technologies Co., Ltd. (Changzhou, China). The raw data files were extracted using extracted by MarkerView 1.3 (AB Sciex, Concord, ON, Canada). The metabolites were identified by BMDB (http://www.rumendb.ca/cgi-bin/browse.cgi (accessed on 29 November 2021)), HMDB (http://www.hmdb.ca/ (accessed on 29 November 2021)), METLIN (https://metlin.scripps.edu/landing_page.php?pgcontent=mainPage (accessed on 29 November 2021)) databases. The principal component analysis (PCA) was conducted on all data which is used to analyze the clustering of the analysis data of each group and remove abnormal samples. Afterwards, a supervised data analysis was carried out on the differences between the divisions, mainly using orthogonal projections to latent structures discriminant analysis (OPLS-DA). PCA and OPLS-DA analysis were conducted on Metaboanalyst 3.0 (Montréal, QC, Canada). CytoScape 3.4.0. (Boston, MA, USA) was used to draw the network mapping. The differentially expressed metabolites were screened with a criterion of variable importance in projection (VIP) >1 and *p* < 0.05.

### 4.7. Statistical Analysis

GraphPad Prism 8.0 (La Jolla, CA, USA) was applied in the present study to perform the data analysis. Significant differences among treatments were assessed using one-way ANOVA test followed by Tukey’s multiple comparisons, which represents as *p* < 0.05.

## Figures and Tables

**Figure 1 toxins-14-00141-f001:**
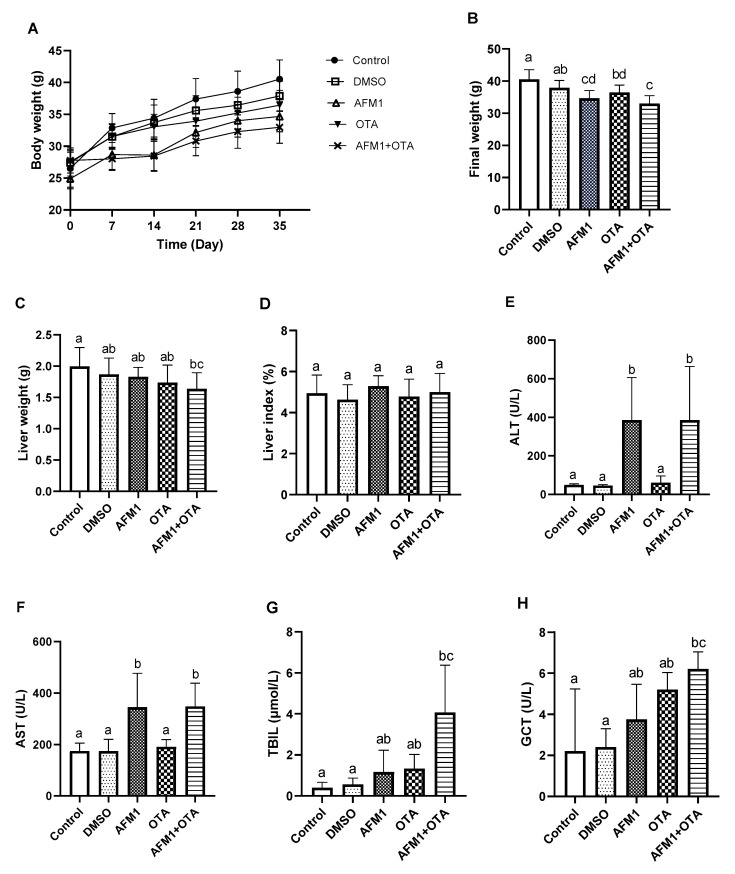
Effects of individual and combined AFM1 and OTA on body weight, liver index, and serum biochemical indicators of mice. (**A**) body weight-time growth curve, (**B**) final body weight, (**C**) liver weight, (**D**) liver index, (**E**) alanine aminotransferase (ALT), (**F**) aspartate aminotransferase (AST), (**G**) total bilirubin (TBIL), and (**H**) glutamyltransferase (GCT). Results are represented as mean ± SD (*n* = 12). Different letters (a,b,c,d) indicate significant differences (*p* < 0.05). Control represents the untreated group, DMSO represents the DMSO vehicle group, AFM1 represents AFM1 at 3.5 mg/kg b.w., OTA represents OTA at 3.5 mg/kg b.w., and AFM1+OTA represents the combined AFM1 and OTA group (3.5 mg/kg b.w. + 3.5 mg/kg b.w.).

**Figure 2 toxins-14-00141-f002:**
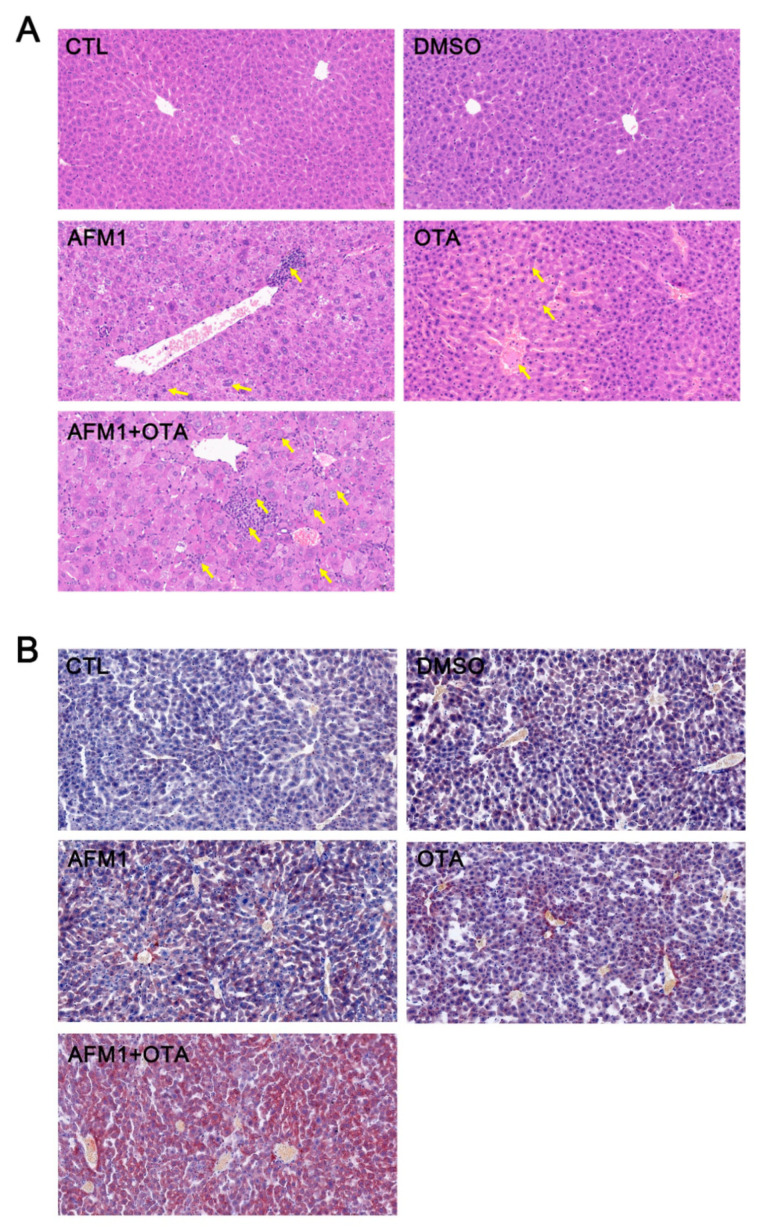
Histopathological examination of control, DMSO, AFM1, OTA, and AFM1+OTA-treated liver tissues by (**A**) hematoxylin and eosin (HE) (400×) staining, (**B**) oil red O (200×) and (**C**) the percentage of lipid droplet area. Different letters (a,b) indicate significant differences (*p* < 0.05). CTL represents the untreated group, DMSO represents the DMSO vehicle group, AFM1 represents AFM1 at 3.5 mg/kg b.w., OTA represents OTA at 3.5 mg/kg b.w., and AFM1+OTA represents the combined AFM1 and OTA group (3.5 mg/kg b.w. + 3.5 mg/kg b.w.).

**Figure 3 toxins-14-00141-f003:**
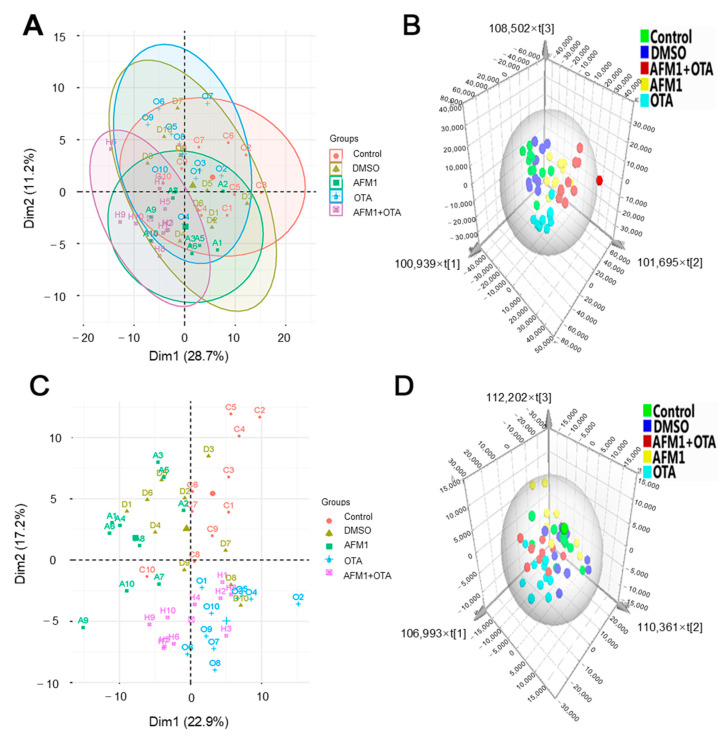
PCA score plots and OPLS-DA analysis of liver and serum samples. (**A**) The scores plot of PCA in the liver, (**B**) the scores plot of PLS-DA in the liver, (**C**) the scores plot of PCA in serum, (**D**) the scores plot of PLS-DA in serum. C represents the untreated group (Control), D represents the DMSO vehicle group (DMSO), A represents AFM1 at 3.5 mg/kg b.w. (AFM1), O represents OTA at 3.5 mg/kg b.w. (OTA), and H represents the combined AFM1 and OTA group (3.5 mg/kg b.w. + 3.5 mg/kg b.w.) (AFM1+OTA).

**Figure 4 toxins-14-00141-f004:**
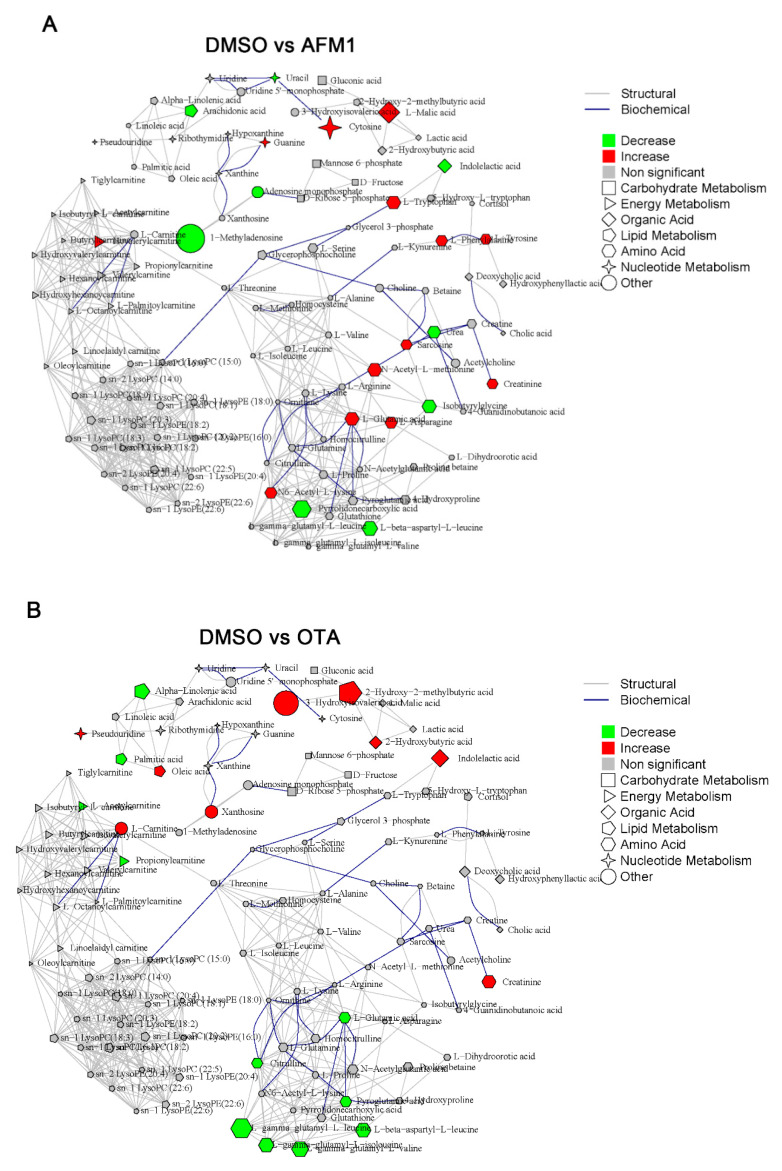
Metabolite integrated metabolic pathway analysis in liver of mice treated with individual and combined AFM1 and OTA. (**A**) DMSO vs. AFM1, (**B**) DMSO vs. OTA, (**C**) DMSO vs. AFM1+OTA, and (**D**) the number of changed metabolites in different mycotoxins treatment. Red represents the upregulated metabolites and the green represents the downregulated metabolites with criterion of FC > 1.50, and *p* < 0.05.

**Figure 5 toxins-14-00141-f005:**
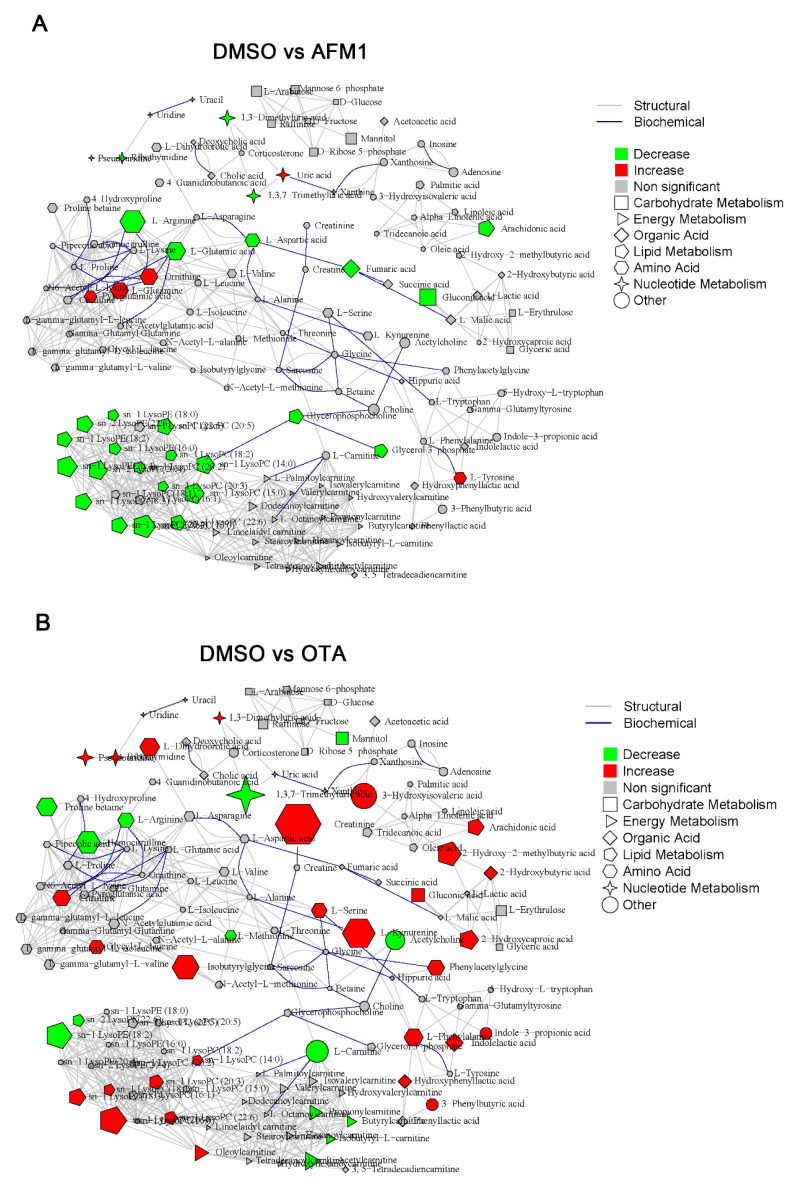
Metabolite integrated metabolic pathway analysis in serum of mice treated with individual and combined AFM1 and OTA. (**A**) DMSO vs. AFM1, (**B**) DMSO vs. OTA, (**C**) DMSO vs. AFM1+OTA and (**D**) the number of changed metabolites in different mycotoxins treatment. Red represents the upregulated metabolites and green represents the downregulated metabolites with criterion of FC > 1.50, and *p* < 0.05.

**Figure 6 toxins-14-00141-f006:**
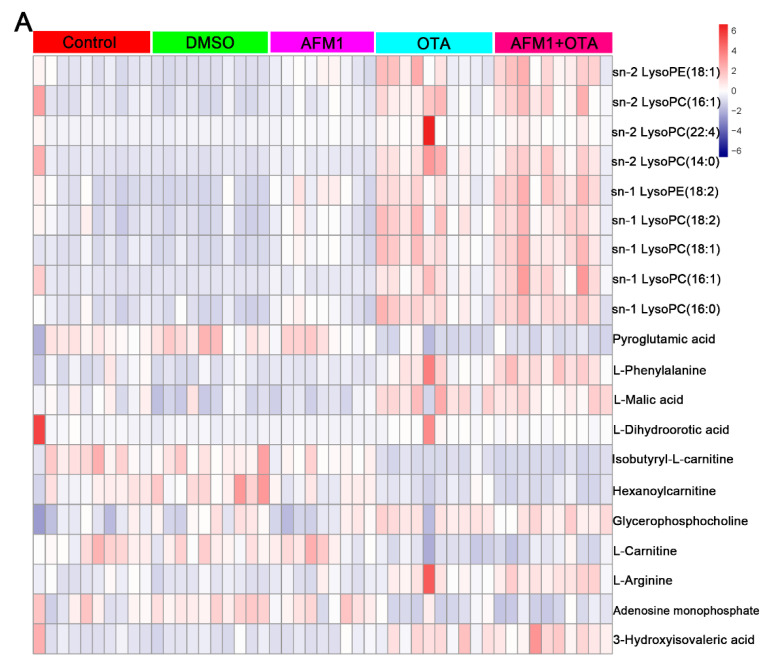
Heat map representation of metabolites mice under different conditions of mycotoxins treatment. (**A**) liver tissue and (**B**) serum samples. Red and blue represent higher and lower level of the respective metabolites.

**Figure 7 toxins-14-00141-f007:**
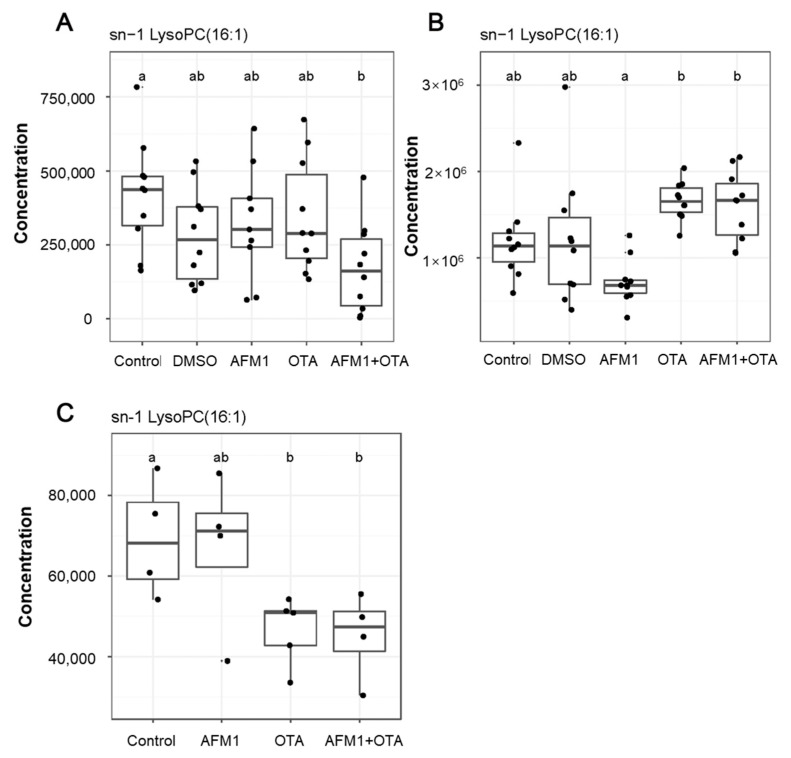
Boxplots of AFM1, OTA and AFM1+OTA regulation of co-differential metabolites in mice (**A**) liver and (**B**) serum as well as (**C**) HepG2 cell.

## Data Availability

Data are available upon request; please contact the contributing authors.

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
