# Peer review of "Metabolomic Analysis Reveals the Mechanisms of Hepatotoxicity Induced by Aflatoxin M1 and Ochratoxin A"

_toxins, 2022, doi:10.3390/toxins14020141_

Round 1

Reviewer 1 Report

the work is interesting. Not extrapolated to humans. It seems to me lot of significance for much deviatión. They should have done more experiments.

Can you explain the deviations and their significance?

Author Response

Dear reviewer,

Thank you very much for providing an opportunity for us to revise our paper, entitled “Metabolomic analysis reveals the mechanisms of hepatotoxicity induced by aflatoxin M1 and ochratoxin A” (Manuscript toxins-1507589) for your journal. And thank you so much for the constructive comments and valuable suggestions on the manuscript. According to the valuable advices, we have made a minor revision to this manuscript.

We hope that the manuscript could be acceptable for publication in your journal.

Responds to reviewers' comments:

Reviewer #1:

The work is interesting. Not extrapolated to humans. It seems to me lot of significance for much deviatión. They should have done more experiments.

Can you explain the deviations and their significance?

AU: Thanks for your suggestions. As your suggestion, firstly, we have deleted the related contents about humans’ exposure to mycotoxins in the revised manuscript.

        Secondly, it is actually important to explore the underlying mechanisms that why significant deviations were existed in individual and combined AFM1 and OTA treatment, and we will to conduct more in-depth researches as your suggestion. The possible reasons summarized by studies have been added in the revised manuscript at line 286-290 as ‘The possible mechanisms of different effects on oxidative damage caused by individual AFM1 and OTA were reported as previous study [56]. The lipophilic structure of these two toxins and their competition for glutathione in cells could partly explained the deviations of cytotoxicity induced by AFM1 and OTA, and further studies are needed to research the exact mechanisms’.

        Thirdly, we added the related contents about the significance of the present study at line 303-308 of the revised manuscript as ‘This finding suggested that although the carcinogen classification level of AFM1 (Group 1) is higher than that of OTA (Group 2B), the adverse effects of OTA cannot be ignored. And this finding also provided a basis that OTA should be classified as a higher level of human carcinogen as previous studies demonstrated [10,11]. In addition, the limit standard for OTA in milk should also be considered to establish due to its higher hepatoxicity than AFM1’.

Reviewer 2 Report

An excellent and comprehensive manuscript with a clear goal.

In this study, the phenotypic results and metabolomics analysis in vivo demonstrated that the combination of AFM1 and OTA performed more serious adverse effects on mice liver, displaying the synergistic effect, in which OTA performed the dominant role. Namely, by means of the combined metabolomics analysis in vitro and in vivo, authors found that LysoPCs, more especially, LysoPC (16:1) were the main type of metabolites affected by AFM1 and OTA, which are related to oxidative stress, resulting in hepatotoxicity.

In general, the manuscript is well written and the reviewer found just some technical errors in the manuscript (please see attached file).

Therefore, please check English spelling and grammar once again throughout the whole manuscript!

All research activities were performed in detail.

Statistical processing of the obtained results and their categorization were also systematically performed.

Material and methods: Very well and clearly written.

Results: This section is very nicely written with all the necessary and concise accompanying explanations.

The results correspond to the objectives of the study.

The figures are in a satisfactory resolution so that all the mentioned and explained details are clearly visible.

Discussion: The discussion part is excellent and explained to a completely satisfactory extent.

The references used are carefully selected and also up-to-date.

Author Response

Dear reviewer,

Thank you very much for providing an opportunity for us to revise our paper, entitled “Metabolomic analysis reveals the mechanisms of hepatotoxicity induced by aflatoxin M1 and ochratoxin A” (Manuscript toxins-1507589) for your journal. And thank you so much for the constructive comments and valuable suggestions on the manuscript. According to the valuable advices, we have made a revision to this manuscript.

We hope that the manuscript could be acceptable for publication in your journal.

Responds to reviewers' comments:

Reviewer #2:

An excellent and comprehensive manuscript with a clear goal.

In this study, the phenotypic results and metabolomics analysis in vivo demonstrated that the combination of AFM1 and OTA performed more serious adverse effects on mice liver, displaying the synergistic effect, in which OTA performed the dominant role. Namely, by means of the combined metabolomics analysis in vitro and in vivo, authors found that LysoPCs, more especially, LysoPC (16:1) were the main type of metabolites affected by AFM1 and OTA, which are related to oxidative stress, resulting in hepatotoxicity.

In general, the manuscript is well written and the reviewer found just some technical errors in the manuscript (please see attached file).

Therefore, please check English spelling and grammar once again throughout the whole manuscript!

All research activities were performed in detail.

Statistical processing of the obtained results and their categorization were also systematically performed.

Material and methods: Very well and clearly written.

Results: This section is very nicely written with all the necessary and concise accompanying explanations.

The results correspond to the objectives of the study.

The figures are in a satisfactory resolution so that all the mentioned and explained details are clearly visible.

Discussion: The discussion part is excellent and explained to a completely satisfactory extent.

The references used are carefully selected and also up-to-date.

AU: Thanks for your suggestion. As your suggestion, we have modified the English grammar as the attached file. In addition, we also have checked English spelling and grammar throughout the whole manuscript and modified them in the revised manuscript.

Reviewer 3 Report

In this paper, the authors reveal the mechanisums of hepatotoxicity induced by combined aflatoxin M1 and ochratoxin A. This paper is fairly well-written, and could be accepted for publication after a minor revision.

Specific comments

  1. The toxicology of aflatoxin M1 and ochratoxin A had been reported by other researches. The authors should compare their results with previous reports.
  2. The figures in the supplementary materials was not in good conditions. The authors should provide images with high resolutions in the revised supplementary materials.

Author Response

Dear reviewers,

Thank you very much for providing an opportunity for us to revise our paper, entitled “Metabolomic analysis reveals the mechanisms of hepatotoxicity induced by aflatoxin M1 and ochratoxin A” (Manuscript toxins-1507589) for your journal. And thank you so much for the constructive comments and valuable suggestions on the manuscript. According to the valuable advices, we have made a revision to this manuscript.

We hope that the manuscript could be acceptable for publication in your journal.

Responds to reviewers' comments:

Reviewer #3:

In this paper, the authors reveal the mechanisms of hepatotoxicity induced by combined aflatoxin M1 and ochratoxin A. This paper is fairly well-written, and could be accepted for publication after a minor revision.

Specific comments

  1. The toxicology of aflatoxin M1 and ochratoxin A had been reported by other researches. The authors should compare their results with previous reports.

AU: Thanks for your suggestion. As your suggestion, we added the related contents in line 291-299 of the revised manuscript as ‘Except hepatotoxicity, the toxicology induced by AFM1 and OTA also had been reported by previous studies. Metabolomics analysis showed that when AFM1 and OTA were combined together, OTA exhibited the dominant effect on the alteration of kidney metabolic processes [56], which was consistent with the role of OTA in the present study. The synergistic effect of the combined AFM1 and OTA was demonstrated in disrupting intestinal integrity, including the decreased cell viability and the expression of tight junction proteins as well as mucins, increased epithelial permeability and intestinal inflammation [57-61]. And the synergy was consistent with the combined effect on hepatotoxicity induced by AFM1 and OTA in the present study’.

  1. The figures in the supplementary materials was not in good conditions. The authors should provide images with high resolutions in the revised supplementary materials.

AU: Thanks for your suggestion. As your suggestion, we have modified the figures of supplementary materials in the revised manuscript.
